# Metabolic Profiling Analysis Reveals the Potential Contribution of Barley Sprouts against Oxidative Stress and Related Liver Cell Damage in Habitual Alcohol Drinkers

**DOI:** 10.3390/antiox10030459

**Published:** 2021-03-15

**Authors:** Hyerin Park, Eunok Lee, Yunsoo Kim, Hye Yoon Jung, Kwang-Min Kim, Oran Kwon

**Affiliations:** 1Department of Nutritional Science and Food Management, Ewha Womans University, Seoul 03760, Korea; rina2129@naver.com (H.P.); sookim726@gmail.com (Y.K.); hy1630@hanmail.net (H.Y.J.); 2Department of Nutritional Science and Food Management, Graduate Program in System Health Science and Engineering, Ewha Womans University, Seoul 03760, Korea; flffl0204@naver.com; 3Department of Family Practice and Community Health, Ajou University School of Medicine, Suwon 16449, Korea; gaksi@ajou.ac.kr

**Keywords:** barley sprout, alcoholic fatty liver, oxidative stress

## Abstract

Chronic excessive alcohol consumption is associated with multiple liver defects, such as steatosis and cirrhosis, mainly attributable to excessive reactive oxygen species (ROS) production. Barley sprouts (*Hordeum vulgare* L.) contain high levels of polyphenols that may serve as potential antioxidants. This study aimed to investigate whether barley sprouts extract powder (BSE) relieves alcohol-induced oxidative stress and related hepatic damages in habitual alcohol drinkers with fatty liver. In a 12-week randomized controlled trial with two arms (placebo or 480 mg/day BSE; *n* = 76), we measured clinical markers and metabolites at the baseline and endpoint to understand the complex molecular mechanisms. BSE supplementation reduced the magnitude of ROS generation and lipid peroxidation and improved the glutathione antioxidant system. Subsequent metabolomic analysis identified alterations in glutathione metabolism, amino acid metabolism, and fatty acid synthesis pathways, confirming the role of BSE in glutathione-related lipid metabolism. Finally, the unsupervised machine learning algorithm indicated that subjects with lower glutathione reductase at the baseline were responders for liver fat content, and those with higher fatigue and lipid oxidation were responders for γ-glutamyl transferase. These findings suggest that BSE administration may protect against hepatic injury by reducing oxidative stress and changing the metabolism in habitual alcohol drinkers with fatty liver.

## 1. Introduction

Alcohol-related deaths account for 5.3% of all deaths worldwide [1]. Approximately 25% of deaths attributable to alcohol consumption are due to alcoholic liver disease [2]. Many factors are involved in the pathogenesis of alcoholic liver disease, including alcohol metabolism-associated oxidative stress and abnormal glutathione (GSH) metabolism [3]. The liver is the primary site where alcohol is oxidized to acetaldehyde by alcohol dehydrogenase (ADH) in the cytosol, cytochrome P450 2E1 (CYP2E1) in the microsome, and catalase (CAT) in the peroxisomes [4]. ADH-mediated alcohol metabolism generates the reduced form of nicotinamide adenine dinucleotide, promoting steatosis by stimulating fatty acid synthesis. Moreover, increased expression of CYP2E1 generates excessive reactive oxygen species (ROS) [5], leading to the peroxidation of membrane lipids and proteins, inflammation, and insulin resistance, which may, in turn, favor the development of hepatic steatosis [6,7,8,9,10]. Alcohol-induced excessive ROS generation may also enhance γ-glutamyl transferase (GGT) release from the damaged hepatocellular membrane [11]. Thus, enhanced GGT release has been widely accepted as an indicator of alcohol-induced hepatocyte injury [12]. Although the underlying mechanisms remain unclear, the link between alcoholic fatty liver, oxidative stress, and hepatic enzymes’ breakdown has been proposed in animals and humans [13,14].

Barley (*Hordeum vulgare* L.) has been cultivated as a food crop since ancient times. Recently, barley sprouts, the young leaves of barley harvested around 10 days after seeding, have begun to receive much attention as a possible food supplement due to its beneficial effects on oxidative stress, in vitro and in vivo [15], as well as on atherosclerosis [16] and depression [17]. Barley sprouts are characterized by a high level of saponarin (apigenin-6-C-glucosyl-7-*O*-glucoside), a potent antioxidant in the flavonoid family [18]. Saponarin exerted antioxidant effects against cocaine or paracetamol-induced hepatotoxicity in rats [19,20]. In another animal study, Lee et al. [21] reported the protective effect of barley sprouts against alcohol-induced liver injury. However, barley sprouts’ protective effect against alcoholic liver injury has not been elucidated in humans.

Therefore, in this study, we hypothesized that barley sprouts might help relieve alcohol-induced oxidative stress and related liver cell damage. To test this hypothesis, we measured the effects of barley sprouts versus a placebo on antioxidant capacities, liver enzymes, and liver fat content in habitual alcohol drinkers with fatty liver in a randomized, controlled clinical trial. Furthermore, we performed gas chromatography coupled to time-of-flight mass spectrometry (GC-TOF-MS) for the untargeted metabolomics in plasma to explore the underlying mechanisms.

## 2. Materials and Methods

### 2.1. Study Materials

The test and placebo materials were provided by Novarex, Inc. (Cheongju, Korea). Briefly, dried barley sprouts were extracted with 30% aqueous food-grade ethanol at 20 °C, filtered, and centrifuged. The supernatant was concentrated to 35° Brix and then spray-dried with dextrin to obtain the barley sprouts extract powder (BSE). We first analyzed the BSE using UHPLC-LTQ-Orbitrap-MS/MS and identified a total of 31 chemical fingerprints (Figure 1), including 17 polyphenols, 4 polyamines, and 10 lipids (Appendix A). Then, the BSE was standardized with saponarin at a concentration of 15 mg/g using a high-performance liquid chromatography (HPLC) apparatus equipped with a UV detector (1260 Infinity, Agilent Technologies, Palo Alto, CA, USA) and a Capcell Pak C18 column (250 mm × 4.6 mm, 5 μm; Shiseido, Tokyo, Japan). The 68.57% BSE, 22.43% maltodextrin, 7% caramel color, 1% silicon dioxide, and 1% magnesium stearate mixture was packed in a capsule to provide 240 mg as BSE. For the placebo capsule, BSE was replaced by weight with dextrin.

### 2.2. Subjects

Subjects were recruited from Ajou University Medical Center (Suwon, Korea). The inclusion criteria were as follows: (1) habitual alcohol consumption over the previous year (more than 140 g/week for males and 70 g/week for females); (2) aspartate aminotransferase (AST) greater than 40 U/L for males and 32 U/L for females, a ratio of AST to alanine aminotransferase (ALT) greater than 1, or GGT greater than 61 U/L for males and 36 U/L for females; and (3) the presence of fatty liver diagnosed by ultrasonography. The exclusion criteria were as follows: (1) alcohol overuse (more than 420 g/week); (2) one of more of AST, ALT, GGT, and total bilirubin levels more than three times over the standard upper limit; (3) presence or history of disease (liver diseases, including viral hepatitis; primary biliary cirrhosis; liver disease induced by drug or operation; inherited liver disease; infectious disease, including tuberculosis; severe cardiovascular disease; digestive system disease, including inflammatory bowel disease; renal disease; thyroid disease; autoimmune disease; malignant tumor; multisystem failure; serious mental illness; and HIV-positive); (4) liver or bone marrow transplantation; (5) continuous consumption of sprout products within the preceding 2 weeks; (6) taking medication, oriental medicine, and/or dietary supplements that may affect liver function and antioxidant/anti-inflammatory properties within the preceding 1 month; (7) hypersensitivity to barley sprout or any ingredients in the study product; (8) participation in another clinical trial within the preceding 1 month; and (9) pregnancy or lactation.

All subjects gave their informed consent for inclusion before they participated in the study. The study protocol was reviewed and approved by the Institutional Review Board (IRB) of Ajou Medical Center (No. AJIRB-BMR-GEN-16-499), and the study was conducted following the Declaration of Helsinki. The study protocol was also registered on the International Clinical Trials Registry Platform of the WHO on March 20, 2017, with the following identification number: KCT0002891.

### 2.3. Study Design

The study was designed as a randomized, double-blinded, paralleled, placebo-controlled trial. After screening and obtaining written informed consent, a total of 76 eligible subjects were randomly assigned either the placebo or BSE for 12 weeks. Randomization was performed using computer-generated random numbers, and the group allocation was blinded for both the investigators and participants. The safe daily intake was estimated to be 480 mg/day of BSE using the result obtained from a previous study of mice (100 mg/kg BW/day) [21] and a conversion factor of 0.08, according to the US FDA guidance on determining the estimated maximum recommended starting dose for initial clinical trials in adult healthy volunteers [22]. The subjects were advised to take two capsules daily with enough water. During the trial, subjects were instructed to maintain their usual diet and lifestyle but refrain from eating or drinking sprout foods and beverages. Dietary intakes, recommended food scores (RFSs), responses to the International Physical Activity Questionnaire (IPAQ), and fatigue severity score (FSS) were recorded at baseline and weeks 6 and 12. Fasting blood and urine samples were collected at baseline and week 12 to determine the biochemical parameters and metabolites.

### 2.4. Liver Fat Content by Magnetic Resonance Imaging (MRI)

Liver fat content was measured by MRI using a 3.0-T MRI system (GE Healthcare, Milwaukee, WI, USA). MRI proton density fat fraction (PDFF) estimates were measured with confounder-corrected chemical shift-encoded (CSE) MRI techniques to avoid the effects of variations in field strength. Quantitative CSE-magnetic resonance (MR) images were acquired with a multi-echo, three-dimensional, spoiled gradient-echo sequence, to generate confounder-corrected PDFF maps over the entire liver. The acquisition parameters at 3 T generally included a repetition time (TR) of 8.6 ms; 6 echoes (initial echo time [TE] = 1.2 ms; ΔTE = 1.0 ms) acquired in two interleaved echo trains of 3 echoes per TR; 44 × 44 cm field of vision (FOV); 256 × 160 matrix; 8-mm slice thickness; 32 slices; 3° flip angle; and ±111-kHz receiver bandwidth. The PDFF MR images were calibrated to allow the direct region of interest (ROI) placement to derive an estimate of the PDFF, expressed as an absolute percentage (0–100%).

### 2.5. Biochemical Measurements

ALT, AST, and GGT were measured using an automatic analyzer (Cobas 8000 c702 analyzer, Roche Diagnostics, Mannerheim, Germany). Plasma oxidized low-density lipoprotein (ox-LDL) was measured using a sandwich enzyme-linked immunosorbent assay kit according to the manufacturer’s instruction (Mercodia, Uppsala, Sweden). Commercially available colorimetric assay kits were used to assess erythrocyte superoxide dismutase (SOD), CAT, GSH, glutathione peroxidase (GPx), glutathione reductase (GR) (all from Cayman, Ann Arbor, MI, USA), and glutathione-S-transferase (GST) (Abcam, Cambridge, UK). The plasma ROS level was determined by chemiluminescence from luminol (5-amino-2,3-dihydro-1,4-phthalazinedione; Sigma-Aldrich, St. Louis, MO, USA) using a chemiluminescence Fluoroskan Ascent FL (Thermo Fisher Scientific, Vantaa, Finland) at 37 °C [23]. After adjusting for background levels, the ROS area under the curve (AUC) was calculated by the trapezoidal rule.

For the determination of oxidative damage, malondialdehyde (MDA) was quantified in urine using an HPLC instrument equipped with a fluorescence detector (Shiseido, Tokyo, Japan) and a Capcell Pak C18 column (250 mm × 4.6 mm, 5 μm; Shiseido) [24]. The mobile phase was 50 mM phosphate buffer:methanol buffer (7:3, *v*/*v*, pH 6.8) at an isocratic flow rate of 1 mL/min. A standard curve was constructed using 1,1,3,3-tetra ethoxy propane solutions for calibrating the peak of the MDA–TBA adduct.

### 2.6. Metabolomics Analysis

A chemical fingerprint of the BSE was analyzed by ultrahigh-performance liquid chromatography linear trap quadrupole orbitrap tandem mass spectrometry (UHPLC-LTQ-Orbitrap-MS/MS) equipped with Vanquish binary pump H system (Thermo Fisher Scientific, Waltham, MA, USA) and a Phenomenex Kinetex C18 column (100 mm × 2.1 mm, 1.7 μm; Torrance, CA, USA) [25]. The final concentration was 1000 ppm. The mobile phase consisted of 0.1% formic acid (solvent A) and acetonitrile containing 0.1% formic acid (solvent B). The gradient was 5% solvent B for 1 min, and linearly increased from 5% to 100% over 9 min, and then decreased to 5% for 2 min. Mass spectra were acquired over the range of 100–1500 *m/z* in the electrospray ionization (ESI) modes. The mass spectrometer was operated under the following conditions: source voltage 5 kV, capillary voltage 45 kV, capillary temperature 275 °C, and source temperature 80 °C.

Plasma samples were centrifuged at 13,000 *g* at 4 °C for 10 min after mixing with methanol at −20 °C for 1 h. The supernatant was filtered and dried using a speed vacuum concentrator (Modulspin 31; Biotron, Incheon, Korea). The dried extract was oximated with methoxyamine hydrochloride in pyridine at 30 °C for 90 min and derivatized with *N*-methyl-*N*-trimethylsilyl-trifluoroacetamide at 37 °C for 30 min. The GC-TOF-MS analysis was performed using an Agilent 7890 gas chromatography system (Agilent Technologies, Palo Alto, CA, USA) coupled with an Agilent L-PAL3 autosampler and equipped with a Pegasus High-Throughput-TOF-MS (LECO, St. Joseph, MI, USA) system. An Rtx-5MS column (30 m × 0.25 mm, 0.25 μm; Restek, Bellefonte, PA, USA) was used with a constant flow of 1.5 mL/min of helium as a carrier gas. Samples were injected into the gas chromatograph with the splitless mode. The oven temperature was maintained at 75 °C for 2 min and then ramped at 15 °C/min to 300 °C and held for 3 min. The temperatures of the front inlet and transfer lines were 250 and 240 °C, respectively. The electron ionization was carried out at −70 eV, and the fullscan data were acquired over a range of 50–1000 *m/z*.

### 2.7. Statistical Analysis

The sample size was estimated to be 38 subjects per group to provide an 80% power of demonstrating a significant difference in the AST/ALT ratio based on the previous study [26] and a drop-out rate of 25%. According to the intention-to-treat (ITT) principle, all data were analyzed and tested for normal distribution graphically by evaluating quantile–quantile plots. Potential outliers were identified as observations exceeding 1.5 times the interquartile range. Differences in baseline characteristics between the two groups were assessed using Student’s *t*-test for continuous variables and the Chi-square or Fisher’s exact test for categorical variables. Between or within group differences were analyzed using a linear mixed-effect model, considering a random effect (subject), a random error (within-subject), and fixed effects (treatment, visit, and treatment × visit interaction). The data are expressed as the least square (LS) means ± standard error (SE). The statistical analyses were performed using the Statistical Analysis Systems package version 9.4 (SAS Institute, Cary, NC, USA). Statistical significance was defined as a *p*-value < 0.05. Treatment–subgroup interactions were analyzed by QUalitative INteraction Trees (QUINT) [27] using the R statistical software version 3.6.3 with a QUINT package [28]. After dividing the non-responder versus responder groups based on the selected modifiers, the differences between them were further statistically compared using a linear mixed-effect model.

Multivariate statistical analysis for metabolomics was performed using SIMCA version 16.0.2 software (Umetrics, Inc., Ume, Sweden). We performed orthogonal partial least-squares discriminant analysis (OPLS-DA) to extract the differential metabolites between the BSE and placebo groups. Then, we ranked the relative contribution of the variables to the OPLS-DA model by the variable importance in the projection (VIP) scores. Variables with VIP scores > 1.0 was considered significant features. Differential metabolites were used to investigate the metabolic pathways activated by BSE supplementation using the pathway topology search tool in MetaboAnalyst 4.0 [29] and the Kyoto Encyclopedia of Genes and Genomes (KEGG) database. The false discovery rate (FDR) controlling procedure was performed, and the calculated *q*-value was used to identify the perturbed pathways. The pathways with *q* < 0.1 were considered significant. Differential metabolites were plotted in a heatmap, and the Euclidean distance metric was used for hierarchical clustering using MetaboAnalyst 4.0. 

## 3. Results

### 3.1. Baseline Characteristics of Subjects

The CONSORT flow diagram is shown in Figure 2. Of the 116 subjects recruited, 76 eligible subjects were enrolled and randomized into either the BSE or placebo group. Ten subjects (seven from the placebo and six from the BSE group) were discontinued due to withdrawal of consent (*n* = 5), loss during follow-up (*n* = 2), violation of guidelines (*n* = 1), or investigator’s opinion (*n* =2). There were no significant adverse effects of the BSE or the placebo. Compliance was excellent (the placebo and BSE groups’ average compliance ratios were 95.7 ± 1.1% and 94.4 ± 1.2%, respectively).

The general characteristics of the subjects included in the ITT analysis are shown in Table 1. The study groups were well matched, with no significant difference between groups. Participants were mostly men with a mean age of 48.9 ± 1.0 years, representing habitual alcohol drinkers (>260 g/week) with hepatic damage (ALT 40.6 ± 2.1 IU/L, AST 31.4 ± 1.3 U/L, GGT 96.6 ± 3.9 U/L, and fatty liver). Although the RFSs were statistically significant (19.6 vs. 23.9), they were all classified as the low fruit/vegetable consumption group [30].

### 3.2. Alterations in Alcohol-Induced Oxidative Stress by BSE Supplementation

This study aims to determine the potential role of BSE in relieving alcohol-induced oxidative stress in relation to liver cell damage. Therefore, we measured the oxidative stress levels and antioxidant protective activities in erythrocytes at the baseline and endpoint (Table 2). The oxidative stress level was assessed by determining ROS using luminol-dependent chemiluminescence. There was a significant decrease in the ROS levels from 2254 ± 245 to 1457 ± 245 in the BSE group (*p* = 0.013) accompanied by no change in the placebo group, demonstrating marginally significant interactions between treatment and visits (ß = −805, *p* = 0.072). Oxidant damage levels were assessed by determining urinary MDA and plasma ox-LDL. Although the ox-LDL decreased remarkably in the BSE group, there was almost no change in the placebo group. In contrast, urinary MDA decreased remarkably from 2.43 ± 0.18 to 2.05 ± 0.19 in the BSE group, while it increased from 1.94 ± 0.18 to 2.39 ± 0.19 in the placebo group. As a result, we found that interactions between treatment and visits were negative and statistically significant (ß = −0.82, *p* = 0.026), indicating a significant effect of BSE supplementation on reducing urinary MDA.

Next, antioxidant protective enzymes and GSH systems were investigated in erythrocytes to determine the protective effects of BSE against oxidative stress. As expected, there were no significant changes in SOD, CAT, and GPx levels. However, the GSH system seemed to be selectively improved by showing a significant increase in the GST level in the BSE group compared with the placebo group (ß = 20.9, *p* = 0.039), with no significant changes in the GSH/oxidized glutathione (GSSG) and GR level.

### 3.3. Alterations in Plasma Metabolomic Profile by BSE Supplementation

We analyzed plasma samples by GC-TOF-MS to compare the metabolic features before and after BSE supplementation. The metabolomics analysis identified 19,459 metabolites in total. The OPLS-DA score was calculated for each group. Separation between the samples before and after supplementation was evident in the OPLS-DA plots (Figure 3A; R^2^X = 0.294, R^2^Y = 0.987, Q^2^ = 0.221). To further illustrate the BSE-induced alterations in the circulating metabolome, heatmap analyses were performed. Consistent with the OPLS-DA score plot, the heatmap demonstrated differential levels of 16 metabolites before and after BSE supplementation (Figure 3B).

The KEGG metabolic pathway analyses enabled to determine which metabolic pathways were impacted by BSE supplementation. Figure 4 shows that BSE supplementation significantly impacted three metabolic pathways: GSH metabolism; alanine, aspartate, and glutamate metabolism; and fatty acid synthesis. In GSH metabolism, pyroglutamic acid and glycine tended to decrease (*q* = 0.076 and 0.085), and cysteine was decreased significantly (*q* = 0.043) after 12-week BSE supplementation compared with the initial level. In alanine, aspartate, and glutamate metabolism, aspartic acid tended to decrease (*q* = 0.085), and asparagine, alanine, and propionic acid were decreased significantly (*q =* 0.011, 0.043, and 0.011, respectively) after BSE supplementation. In fatty acid synthesis, although palmitic acid tended to decrease (*q =* 0.076), there were significant decreases in the acetate (*q* = 0.025), decanoic acid (*q* = 0.010), dodecanoic acid (*q* = 0.025), and stearic acid (*q* = 0.010) levels after BSE supplementation relative to the initial values.

### 3.4. Impacts of BSE Supplementation on Alcohol-Induced Liver Cell Damages

In line with the metabolomics results, the BSE group showed significant improvements in liver fat content from 14.3 ± 1.1% to 12.4 ± 1.1% (*p* < 0.001 vs. baseline) and GGT from 90.5 ± 4.2 to 81.1 ± 4.5 U/L (*p* = 0.025 vs. baseline). Reductions in liver fat content and GGT levels were also noted in the placebo group, although to a relatively lower degree. As a result, interactions between treatment and visits were not statistically significant (Table 3).

Next, we applied QUINT analysis using the treatment effect size criterion to check whether subject characteristics measured at baseline had significant interactions with the BSE supplementation benefits on the reversal of liver cell damage, namely liver fat content and GGT (Figure 5). The result for the pre- to post-supplementation change in liver fat content was a pruned tree with two leaves. The split of the tree involved the variable “GR activity” and a split point of 141.9 nmol/min/mL. The results indicated that the subjects with a baseline GR activity lower or equal to 141.9 nmol/min/mL were responders (Leaf 1, *n* = 28) whose liver fat content was significantly reduced by BSE supplementation compared with the placebo group (*p* = 0.003). For the GGT levels, there were three pruned trees with two leaves individually. The split point of the “FSS” variable was higher than 18.5, that of the “ROS AUC” variable was higher than 539.85, and that of the “ox-LDL” variable was higher than 60.58 μg/mL. The results indicated that the subjects with FSS (Leaf 2, *n* = 51), ROS AUC (Leaf 2, *n* = 54), or ox-LDL (Leaf 2, *n* =51) values higher than the corresponding split points were responders whose GGT levels were remarkably reduced by BSE supplementation compared with the placebo group. However, the responder groups for the FSS (*p* = 0.044) and ox-LDL (*p* = 0.030) subgroups only reached statistical significance for GGT.

## 4. Discussion

Alcohol-induced oxidative stress is likely a powerful mechanism for hepatic cell death and tissue injury [31]. The most convincing data supporting the link between oxidative stress and alcohol-induced liver injury comes from animal studies using the intragastric feeding model. In these studies, the development of alcoholic liver injury was associated with enhanced lipid peroxidation, generation of highly reactive free radicals, and downregulation of the hepatic antioxidant defense system [32]. However, alcohol-induced oxidative stress can be reversed by dietary supplementation. Pari et al. [33] demonstrated in a rat study that treatment with grape leaf extracts reduced alcohol-induced oxidative stress by restoring the enzymic and non-enzymatic antioxidant levels in the liver and kidney. In the current study, we observed that 12-week BSE supplementation reduced the magnitude of ROS generation and lipid peroxidation, respectively, and improved the GSH antioxidant system, hepatic serological biomarkers, and steatosis in habitual alcohol drinkers with fatty liver. To our knowledge, this study is the first human trial to show the effects of BSE on alcohol-induced oxidative stress and related liver cell damage.

We analyzed BSE by UHPLC-LTQ-Orbitrap-MS/MS to gain insight into the potentially active chemical compounds that might contribute to its protective effects against oxidative stress and liver cell damage. As a result, we found that BSE contains a great diversity of glycosylated isovitexin (known as saponarin) and isoorientin (known as lutonarin). Isovitexin and isoorientin are C-glycosides of apigenin and luteolin, respectively, which are dominant metabolites produced during barley grain sprouting. Previous in vitro studies showed that they have even higher antioxidant activities than their corresponding aglycones [15,34], supporting our observations in humans. Several more studies support the hepatoprotective effects of saponarin in preclinical settings. Simeonova et al. [20] demonstrated that saponarin reduced drug-induced oxidative stress and hepatic damage in vitro and in vivo. Therefore, we standardized the BSE with saponarin at a concentration of 15 mg/g.

It is almost impossible in the clinical setting to test the concept of imbalance between ROS production and the antioxidant defense system directly in the liver. Hence, we used circulating samples (plasma, erythrocytes, and urine) because chronic alcohol use leads to oxidative stress markers in the circulatory system and extrahepatic tissues at the same time [32]. The GSH antioxidant system consists of several components, including the GPx, GR, GST, and GSH. Together, these molecules effectively remove hydrogen peroxide [35]. GSH (γ-l-glutamyl-l-cysteinyl glycine) can also interact non-enzymatically with individual ROS to detoxify them [35,36]. Given the considerable amount of harmful oxidants detoxified in a GSH-dependent manner, a compromised GSH antioxidant system has been speculated to be a major contributor to the pathologic liver injury in chronic alcohol drinkers [36]. Consistent with these previous findings, results from the current study indicate that BSE supplementation improved the GSH antioxidant system by selectively targeting the GST level without affecting the GSH level. GST catalyzes the conjugation of GSH with ROS or peroxidized lipids for their elimination from the system [37,38].

Furthermore, we performed a non-targeted GC-TOF-MS metabolomic analysis to explain the possible mechanism of action of BSE in linking the suppression of oxidative stress to the improvement of hepatic cell damages and steatosis. OPLS-DA is a supervised modeling approach for extracting the differential metabolites between the two groups, while PCA is an unsupervised modeling approach for identifying the maximum variance [39]. The OPLS-DA model in the present study displayed greater than 98% goodness-of-fitted, suggesting that the BSE supplementation led to changes in endogenous metabolites. Regarding the link between alcohol-induced oxidative stress and hepatic cell damage, Singh et al. [13] reported higher plasma levels of GGT and MDA and GSH depletion in alcoholics than in healthy volunteers in a clinical setting. GGT is a membrane-bound enzyme that catalyzes the transfer of the γ-glutamyl moiety of GSH to amino acids. GGT is also involved in the synthesis of GSH, playing an important role in the antioxidant defense system [40]. However, an increase in plasma GGT linked to the release of γ-glutamyl amino acids and cysteinyl glycine has been widely used as an index of liver dysfunction or alcohol intake [41], indicating hepatic cell damage and a high turnover rate of hepatic GSH due to oxidative stress [42]. In the current study, BSE supplementation led to significant reductions in pyroglutamic acid, glutamylated amino acids, and GGT in plasma, suggesting that BSE reduced GSH turnover and hepatic cell damage via suppression of oxidative stress. Meanwhile, regarding the link between alcohol-induced oxidative stress and hepatic steatosis, Kalhan et al. [43] reported higher plasma levels of glutamyl peptides and lower plasma levels of GSH in subjects with steatosis than in healthy controls. In the present study, the remarkable reductions in aspartic acid, asparagine, alanine, propionic acid, palmitic acid, acetate, decanoic acid, dodecanoic acid, and stearic acid levels were found after BSE supplementation, implicating the suppression of fatty acid synthesis. Taken together, the alterations in the metabolic response most probably reflect that BSE supplementation restores hepatic function via activating the GSH antioxidant system.

The analysis of hepatic serological biomarkers and steatosis identified that BSE supplementation lowered the liver fat content and GGT levels remarkably in the within-group comparisons, but not in the between-group comparisons. These results prompted us to introduce a robust strategy for identifying the subgroups of subjects for whom the effectiveness of BSE supplementation differs from the others. We used an unsupervised machine learning algorithm, QUINT, which was developed to separate the subjects into responders and non-responders based on an array of variables measured before treatment started in a two-arm clinical trial [28]. Such background characteristics that predict differential effectiveness are called effect modifiers [44]. The analysis revealed that the liver fat content was improved markedly in subjects who initially had relatively lower GR activity. GR is an enzyme that catalyzes the recycling of GSSG to GSH and has a role in the maintenance of GSH homeostasis, especially when the cells are exposed to oxidative stress [45]. On the contrary, GGT improvement was significant in subjects who initially had a relatively higher fatigue or lipid damage level. These findings are important because they indicate the persons who should receive the optimal treatment based on their individual characteristics.

## 5. Conclusions

Here, we revealed that BSE supplementation improves hepatic damages in habitual alcohol drinkers with fatty liver by relieving alcohol-induced oxidative stress and maintaining the GSH antioxidant system. The plasma metabolomics analysis of endogenous metabolites was useful in explaining the link between the suppression of oxidative stress and the improvement in hepatic cell damage and steatosis. We also demonstrated that BSE supplementation might be more effective in subjects who initially had a relatively lower GR activity, higher fatigue level, or more significant oxidative damage in the background. However, we did not explore whether multiple chemical compounds present in BSE exert synergistic actions in protecting against alcohol-induced oxidative stress and hepatic damage. Future studies are needed to understand the complex interactions between multiple BSE components and multiple targets in the body by applying a computational network approach.

## Figures and Tables

**Figure 1 antioxidants-10-00459-f001:**
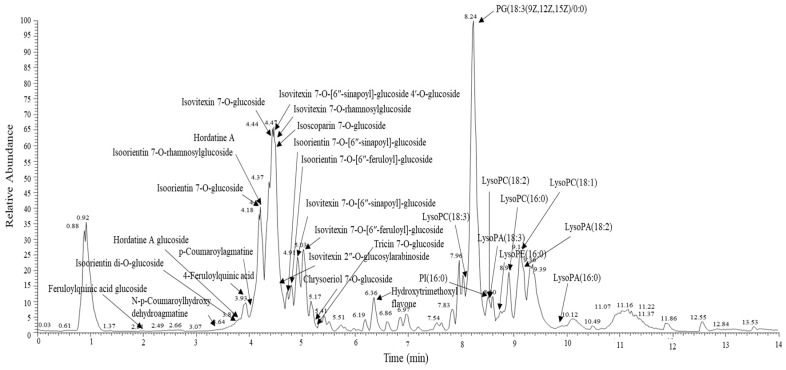
The UHPLC-LTQ-Orbitrap-MS/MS chromatogram of the BSE. UHPLC-LTQ-Orbitrap-MS/MS, ultrahigh-performance liquid chromatography linear trap quadrupole orbitrap tandem mass spectrometry; BSE, barley sprouts extract powder.

**Figure 2 antioxidants-10-00459-f002:**
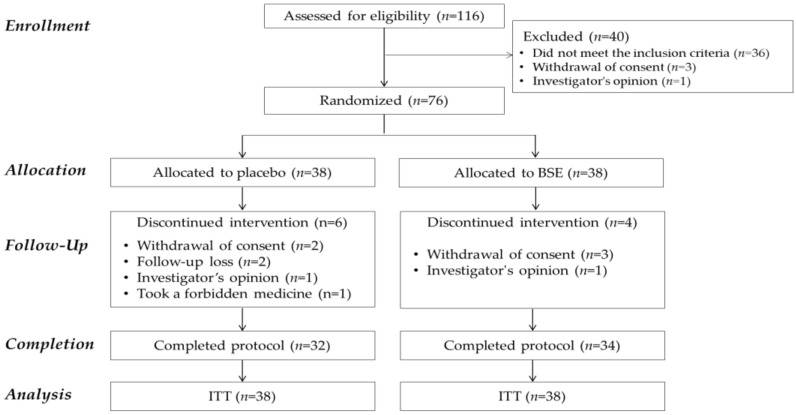
Consolidated standards of reporting trials (CONSORT) diagram of the study. BSE, barley sprouts extract powder; ITT, intention-to-treat.

**Figure 3 antioxidants-10-00459-f003:**
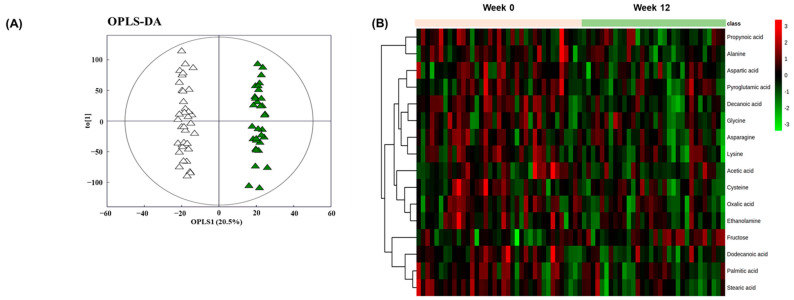
BSE supplementation alters the circulating metabolome. (**A**) OPLS-DA score plot for the GC-TOF-MS data of each sample (△ week 0; ▲ week 12). (**B**) Heatmap overview of the differential metabolomes before and after BSE supplementation. BSE, barley sprouts extract powder; OPLS-DA, orthogonal partial least-squares discriminant analysis; GC-TOF-MS, gas chromatography coupled to time-of-flight mass spectrometry.

**Figure 4 antioxidants-10-00459-f004:**
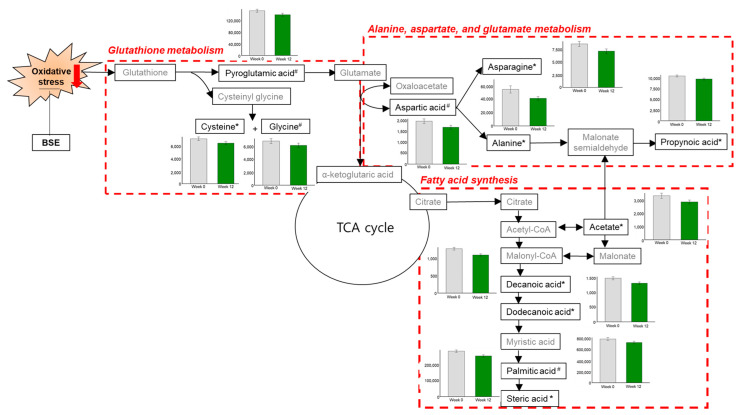
Overview of the KEGG metabolic pathways in response to BSE supplementation. Arrows indicate metabolic direction. Bar graphs compare the individual metabolites’ changes in the BSE group at the baseline (grey) and endpoint (green) of the 12-week study. * *q* < 0.05, # *q* < 0.1. BSE, barley sprouts extract powder.

**Figure 5 antioxidants-10-00459-f005:**
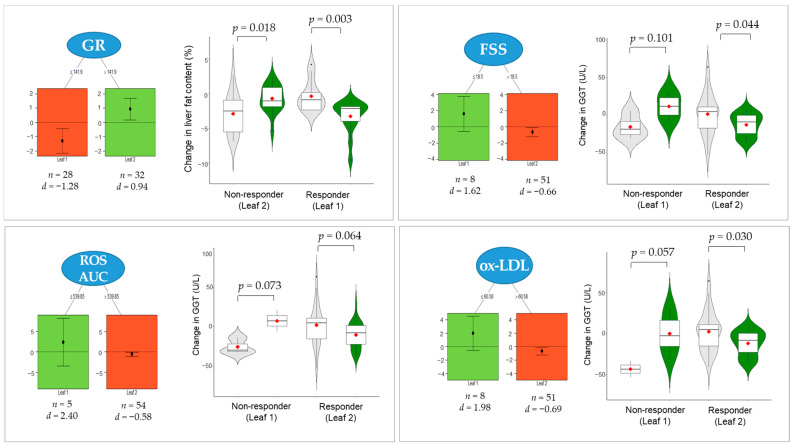
Pruned trees and descriptive statistics of QUINT. The changes (∆0–12 weeks) in liver fat content with moderator variable “GR” and a split point of 141.9 nmol/min/mL (top left). The changes (∆0–12 weeks) in GGT level with moderator variables of “FSS” (a split point of 18.5; top right), “ROS AUC “(a split point of 539.85; bottom left), and “ox-LDL” (a split point of 60.58 μg/mL; bottom right). The corresponding effect sizes *d* are expressed as the standardized mean difference between the group. BSE is more effective for the red leaves than the placebo; the reverse is valid for the green leaves. The *p*-values were assessed using a mixed-model repeated-measures analysis for the treatment × visit interaction terms. Violin plots compare the distribution of the changes in liver fat content and GGT level (
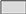
 placebo; 
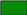
 BSE). The interquartile range and median are shown by the vertical bar and black dash, respectively, and the mean is shown by the red dot. GR, glutathione reductase; FSS, fatigue severity scale; ROS, reactive oxygen species; AUC, area under the curve; ox-LDL, oxidized low-density lipoprotein; GGT, γ-glutamyl transferase; BSE, barley sprout extract powder.

**Table 1 antioxidants-10-00459-t001:** Baseline characteristics of the ITT subjects who participated in this study.

Variable	Placebo (*n* = 38)	BSE (*n* = 38)	*p*-Value
Gender (male/female)	37/1	37/1	1.000
Age (years)	47.6 ± 1.5	50.3 ± 1.5	0.185
Alcohol amount (g/week)	294.3 ± 13.9	264.0 ± 12.9	0.115
Smoker (Y/N)	17/21	14/24	0.484
Smoking amount (cigarettes/day)	16.2 ± 2.5	14.1 ± 1.6	0.473
FSS (total score)	29.0 ± 1.6	30.2 ± 1.8	0.631
Physical activity (MET-min/wk)	2714 ± 612	1845 ± 340	0.220
RFS	19.6 ± 1.2	23.9 ± 1.5	0.028
BMI (kg/m^2^)	27.4 ± 0.4	27.6 ± 0.4	0.671
Liver fat content (%)	15.1 ± 1.5	15.1 ± 1.3	0.984
ALT (IU/L)	42.5 ± 3.2	38.6 ± 2.7	0.357
AST (IU/L)	31.6 ± 2.0	31.2 ± 1.7	0.857
GGT (U/L)	97.7 ± 5.3	95.5 ± 5.7	0.780

Values are presented as the means ± SE or *n*. Student’s *t*-test for continuous variables and chi-square or Fisher’s exact test for categorical variables was used to compare the group differences. ITT, intention-to-treat; BSE, barley sprouts extract powder; FSS, fatigue severity score; RFS, recommended food score; BMI, body mass index; ALT, alanine aminotransferase; AST, aspartate aminotransferase; and GGT, γ-glutamyl transpeptidase.

**Table 2 antioxidants-10-00459-t002:** Oxidative stress markers and antioxidant levels at the baseline and endpoint in the placebo and BSE groups.

Variable	Placebo (*n* = 38)		BSE (*n* = 38)		Estimate	*p*-Value
	Week 0	Week 12	Week 0	Week 12		
ROS AUC	1861 ± 222	1869 ± 241	2254 ± 245	1457 ± 245 *	−805	0.072
Urinary MDA (μmol/g creatine)	1.94 ± 0.18	2.39 ± 0.19	2.43 ± 0.18	2.05 ± 0.19	−0.82	0.026
ox-LDL (μg/mL)	77.3± 2.7	76.5 ± 2.9	72.5 ±2.7	68.4 ± 2.9	−3.3	0.350
SOD/CAT	7.3 ± 1.7	10.1 ± 1.8	9.7 ± 1.8	9.8 ± 1.8	−2.6	0.271
SOD/GPx	0.18 ± 0.02	0.18 ± 0.02	0.21 ± 0.02	0.23 ± 0.02	0.01	0.498
SOD/(CAT+GPx)	0.18 ± 0.02	0.17 ± 0.02	0.20 ± 0.02	0.23 ± 0.02	0.03	0.180
GR (nmol/min/mL)	134.0 ± 7.9	134.9 ± 8.2	139.2 ± 7.8	133.0 ± 8.2	−7.1	0.481
GST (nmol/min/mL)	108.5 ± 6.0	98.4 ± 6.5	100.5 ± 6.0	111.4 ± 6.3	20.9	0.039
GSH/GSSG	4.79 ± 0.39	5.81 ± 0.41	4.25 ± 0.39	4.75 ± 0.41	−0.52	0.351
GSSG/TGSH	0.20 ± 0.01	0.17 ± 0.01	0.21 ± 0.01	0.20 ± 0.01	0.01	0.488

Values are presented as the LS means ± SE. Estimates and *p*-values were obtained using a mixed-model repeated-measures analysis for the treatment × visit interaction terms. * *p* < 0.05, differences within each group. BSE, barley sprouts extract powder; ROS, reactive oxygen species; AUC, area under the curve; MDA, malondialdehyde; ox-LDL, oxidized low-density lipoprotein; SOD, superoxide dismutase; CAT, catalase; GPx, glutathione peroxidase; GR, glutathione reductase; GST, glutathione S-transferase; GSH, reduced glutathione; GSSG, oxidized glutathione; TGSH, total glutathione.

**Table 3 antioxidants-10-00459-t003:** Liver cell damage markers at the baseline and endpoint in the placebo and BSE groups.

Variable	Placebo (*n* = 38)		BSE (*n* = 38)		Estimate	*p*-Value
	Week 0	Week 12	Week 0	Week 12		
Liver fat content (%)	13.0 ± 1.1	11.3 ± 1.1 **	14.3 ± 1.1	12.4 ± 1.1 ***	−0.2	0.725
ALT (IU/L)	37.9 ± 2.2	34.4 ± 2.4	37.8 ± 2.2	35.9 ± 2.3	1.6	0.581
AST (IU/L)	29.2 ± 1.4	29.5 ± 1.4	29.8 ± 1.4	28.9 ± 1.4	−1.2	0.505
AST/ALT ratio	0.78 ± 0.03	0.81 ± 0.04	0.81 ± 0.03	0.81 ± 0.03	−0.04	0.424
GGT (U/L)	92.1 ± 4.2	89.9 ± 4.5	90.5 ± 4.2	81.1 ± 4.5 *	−7.2	0.213

Values are presented as the LS means ± SE. Estimates and *p*-values were obtained using a mixed-model repeated-measures analysis for the treatment × visit interaction terms. * *p* < 0.05, ** *p* < 0.01, *** *p* < 0.001, differences within each group. BSE, barley sprouts extract powder; ALT, alanine aminotransferase; AST, aspartate aminotransferase; GGT, γ-glutamyl transpeptidase.

## Data Availability

Data are included in the article.

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
