# Peer review of "Metabolic Profiling Analysis Reveals the Potential Contribution of Barley Sprouts against Oxidative Stress and Related Liver Cell Damage in Habitual Alcohol Drinkers"

_antioxidants, 2021, doi:10.3390/antiox10030459_

Round 1

Reviewer 1 Report

The manuscript by Park et al. is an interesting study on the role of barley sprouts against oxidative stress and cell damage in habitual alcohol drinkers. The results are very clear and the data well-presented.

However, I have some concerns regarding the basal differences between the experimental groups. On of these concerns, is the fact that urinary MDA is very different between the placebo and the BSE groups. Can you comment on this?

It also appears that the fat content is higher in then BSE group at week 0. Can you present the statistics on this comparison? This could be a problem as the two experimental groups do not present similar basal characteristics.

Another interesting point is the fact that liver fat content also changed significantly in the placebo group from week 0 to week 1. Have you checked if physical activity or food intake changed  during the study period both for the placebo and BSE-treated group?

Author Response

Dear Editor,

We want to thank the editor and reviewers for the time and efforts in providing very detailed and constructive feedback. Comments were precious, directing the manuscript in a way that we had not previously considered.

We have addressed all concerns raised and carefully revised the manuscript to incorporate the reviewer's comments as thoroughly as possible. We have summarized our response below point-by-point, and the corrections have been marked by red color in the revised manuscript.

Kind regards,

Oran Kwon

Reviewer 2 Report

Excessive alcohol consumption is associated with multiple liver defects from steatosis to cirrhosis, mainly attributable to excessive reactive oxygen species (ROS) production. The authors investigated the barley sprouts extract powder (BSE) relieves alcohol-induced oxidative stress and related hepatic damages in habitual alcohol drinkers with fatty liver. The authors also measured clinical markers and metabolites at baseline and endpoint to understand the complex molecular mechanisms. The manuscript was well-prepared with clinic significant. A few minor suggestions were provided as follow:

  1. Although the chemical fingerprints were identified for BSE by UHPLC-LTQ-Orbitrap-MS/MS analysis, but no further discussion or application on the functional components from BSE contributed to prevent the alcohol induced injury.
  2. In Figure 3, why the OPLS-DA was used for the separation? How is PCA or PLS analysis? A discussion or rationale on the reason of using OPLS-DA should be provided.
  3. How was the heatmap were clustered? Which algorithm used? How is the volcano plot for the different changed metabolites?
  4. How were the three metabolic pathways were selected in Figure 4?

Author Response

(The authors gave the same response as above.)

Reviewer 3 Report

This was a well-designed study to translate findings from animal research to the clinical setting of alcohol-induced fatty liver disease. Overall the paper was well written and results were clearly presented. The primary concern of this reviewer was regarding the significant difference in many variables of interest at baseline between the groups, leading to questions regarding the validity of the Quint analysis.

  1. Lines 52,53 “food supplement due to health benefits against in vitro and in vivo oxidative stress [15], atherosclerosis [16], or depression [17]”… this sentence is confusing. Suggest changing to “food supplement due to its beneficial effects on oxidative stress, in vitro and in vivo [15], as well as on atherosclerosis [16] and depression [17]”

  1. Section 2.3 How was the dose scaled from the animal dose? Please provide some details of the calculation or allometric scaling.

  1. Section 3.1. Suggest moving Supplementary Table S1 to this section and removing the text list of compounds on lines 210-223. The list is not informative. Also suggest moving this to the end of the results section consistent with the flow of the discussion.

  1. Table 1 – suggest leaving out the female subject from the analysis as effects of gender difference cannot be assessed with only 1 participant.

  1. The BSE group appears to have significantly different baseline values in a number of parameters as compared with the placebo group. How was that taken into account in the QUINT analysis? The authors should address the potential for bias in measurements, particularly due to the significant difference in ROS at baseline between the groups.

  1. Discussion – The discussion of the BSE analysis (lines 415-428) is at the end of the discussion section whereas these data are at the beginning of the results. Either the data should be moved to the end of the results section (preferred) or this portion should be moved to the top of the discussion section.

  1. Suggest moving Lines 428-432 to the conclusion.

Author Response

(The authors gave the same response as above.)

Round 2

Reviewer 2 Report

All questions had been answered.

Author Response

Thank you. 

Reviewer 3 Report

The authors addressed reviewer comments. There is an error in the first sentence the authors changed that requires editing, as well as reference #31. Otherwise, the manuscript looks good! Thanks for the response!

Author Response

There is an error in the first sentence the authors changed that requires editing, as well as reference #31. 

Reply: Corrected as commented. (Line 53-54, Reference #31) Thank you so much!